# Phytochemical Composition of Different Botanical Parts of *Morus* Species, Health Benefits and Application in Food Industry

**DOI:** 10.3390/plants11020152

**Published:** 2022-01-06

**Authors:** Adriana Ramona Memete, Adrian Vasile Timar, Adrian Nicolae Vuscan, Florina Miere (Groza), Alina Cristiana Venter, Simona Ioana Vicas

**Affiliations:** 1Doctoral School of Biomedical Science, University of Oradea, 410087 Oradea, Romania; adrianamemete@yahoo.com; 2Faculty of Environmental Protection, University of Oradea, 410048 Oradea, Romania; atimar@uoradea.ro (A.V.T.); adyvuscan@yahoo.com (A.N.V.); 3Faculty of Medicine and Pharmacy, University of Oradea, 410073 Oradea, Romania; florinamiere@uoradea.ro (F.M.); alinaventer@gmail.com (A.C.V.)

**Keywords:** *Morus* sp., anthocyanins, flavonols, chlorogenic acid, morusin, health benefits, functional food

## Abstract

In recent years, mulberry has acquired a special importance due to its phytochemical composition and its beneficial effects on human health, including antioxidant, anticancer, antidiabetic and immunomodulatory effects. Botanical parts of *Morus* sp. (fruits, leaves, twigs, roots) are considered a rich source of secondary metabolites. The aim of our study was to highlight the phytochemical profile of each of the botanical parts of Morus tree, their health benefits and applications in food industry with an updated review of literature. Black and white mulberries are characterized in terms of predominant phenolic compounds in correlation with their medical applications. In addition to anthocyanins (mainly cyanidin-3-O-glucoside), black mulberry fruits also contain flavonols and phenolic acids. The leaves are a rich source of flavonols, including quercetin and kaempferol in the glycosylated forms and chlorogenic acid as predominant phenolic acids. Mulberry bark roots and twigs are a source of prenylated flavonoids, predominantly morusin. In this context, the exploitation of mulberry in food industry is reviewed in this paper, in terms of developing novel, functional food with multiple health-promoting effects.

## 1. Introduction

*Morus* is a perennial, woody, economic, fast-growing deciduous tree, distributed worldwide (India, China, Japan, North Africa, Arabia, South Europe) and used for centuries in agriculture, food, cosmetics and medicine due to its chemical composition and pharmacological activity [1,2,3].

There are 11 main species in the world, but the most common species of mulberry are black (*Morus nigra* L.), white (*Morus alba* L.) and red (*Morus rubra* L., also called American mulberry and native to the eastern United States), followed by other species: *Morus australis, Morus bomycis, Morus laevigata, Morus serrata, Morus macroura, Morus cathayana, Morus multicaulis* and *Morus insignis* [4,5,6].

The intensive selection and reproduction of mutations led to the emergence of thousands of varieties, hybrids and polyploids; today, there are 24 species of *Morus* and one subspecies, with at least 100 known varieties, with androgynous flowers, both male or female. Among the varieties with androgynous flowers, pistillate, staminate or even hermaphroditic types can appear [6,7,8].

In general, the mulberry (*Morus nigra* L.) is a diploid plant and has 28 chromosomes (2n = 28), but when analyzing its ploidy, many triploid varieties were also found (2n = (3x) = 42), cultivated for their adaptability, vigorous growth and leaf quality. Tetraploids, pentaploid, hexaploidy and dexoploids varieties, the latter having the highest number of chromosomes (2n = 308), represent one of the most valuable adhesions in row of mulberry genetic resources [9,10].

Mulberry tree is the most important plant belonging to the genus *Morus* of the family Moraceae. It is not a sensitive tree; it is attractive and long lasting and can be planted not only for consumption but also for gardening, edible landscaping, street shade and reducing pollution in the environment and soil. Over thousands of years, it has been domesticated and is now well adapted to grow in almost all types of soil, except swampy, saline or acidic soils [1,11,12].

Mulberry trees can be grown almost throughout the territory, but resistant only where environmental conditions are ensured (light, temperature, water, wind). Light is one of the factors that directly conditions the physiological processes of the mulberry trees, with a preponderant role in its production. The insufficiency of light favors the appearance of mulberry trees pathogens; for this reason, they are planted in sunny areas, and the mulberry rows are placed in an east–west direction. In hilly areas, they are located only on southern, southeastern or southwestern exposures. Being a light-loving tree, for almost all varieties cultivated, a vegetation period of approximately 200 days is required [13,14].

From the point of view of the humidity regime, the mulberry tree is part of the category of mesophilic plants, developing harmoniously on sufficiently moist soils. The content of water from leaf, shoots and roots represents 60–80%, 60–63% and 55–56%, respectively, of the weight of organs. Their water needs are ensured since most of the precipitation is evenly distributed in all the months of the vegetation period, and to a lesser extent, in the groundwater table [4,15,16]. Planting mulberry in nutrient-rich soils with a low acidic or neutral pH will result in the best yield in terms of leaf quality, but also fruit [16,17,18,19].

Asian countries have a long and rich history of large-scale cultivation of white mulberry as a key habitat for silkworm rearing, also known as Sang Shen in China and Oddi in Korea [20]. It is believed that about 5000 years ago, around 2700 B.C., Chinese Empress Si Ling-Shi accidentally discovered that silkworms could be obtained from the cocoon of mulberry leaf worms, and the secret of this art has been kept in China for about 2000 years [4]. On the other hand, in Turkey and Greece, mulberries are mainly grown for fruit production than for foliage [21].

*Morus* species gained interest in the food industry due to their health benefits, being used in the treatment of anemia, tonsillitis and other sore throats, and today, it is supplied in markets in the form of paste called *sangshengao*, used in the preparation of teas with benefits for sight, hearing, kidneys and the liver [22,23].

Herbal preparations are the oldest and richest remedies in human health care, and today, the human being studies, tests and regains confidence in natural, effective remedies that have survived for several generations. According to the World Health Organization, almost eighty percent of the world’s population continues to use complementary therapies, many of which are derived from plants such as mulberry.

Based on the recent literature on mulberry, this study aims to review the effectiveness of mulberry as a whole plant, by highlighting the phytochemical profiles of each botanical part of *Morus* tree and attributing each part to various functional properties, exploring its nutraceuticals applications in medicine and food industries. A systematic update of the existing literature was conducted.

## 2. Research Methodology

Recent studies regarding the phytochemical composition of fruits, leaves, twigs and roots and their beneficial effects for curing different diseases were selected using a PRISMA 2020 flow diagram based on the suggestion of Page et al., 2021 [24]. The steps and selection criteria, followed by the number of the studies used for our review, are shown in the Figure 1. Databases such as PubMed, Scopus, Science Direct, Elsevier, Google Scholar, Google Patents were accessed to search the literature. The Medical Subject Headings keywords included in the search were: “*Morus*” “*Morus nigra*”, “*Morus alba*”, “nutrients *Morus*”, “bioactive compounds *Morus*”, “phytochemicals *Morus*”, “antioxidant capacity *Morus*”, “*Morus* diabetes”, “anticancer *Morus*”, “immunomodulatory *Morus*”, “food industry *Morus*” etc. All information systematized in the tables was obtained from research articles (in vivo/ex vivo/in vitro studies) between 2016 to 2021. Studies published in languages other than English were excluded, except one patent which was written in Romanian. A total of 199 studies were selected and included in this review.

## 3. Nutritional Values of the Botanical Parts of *Morus* sp.

*Morus* tree is considered a medicinal plant. All of its botanical parts (fruits, leaves, twigs, root bark) have their own special effects on the human body, which is why they are used in traditional Chinese and Indian medicine [1,2,23].

### 3.1. Mulberry Fruits

There are multiple *Morus* sp. fruits (about 2 to 3 cm long); they form together and are arranged longitudinally around the central axis, similar to blackberries, and are low in calories, but rich in nutrients and antioxidants, so they can ensure good overall health [25]. The fruits contain a high-water content (over 70%), and the pH values differ between species: *M. alba* presents the highest value (5.6), while the *M. rubra* and *M. nigra* have values of 4.04 and 3.52, respectively [21]. Based on values of pH, total soluble solid and total dry weight, the *M. alba* could be recommended for processing, while *M nigra* may be recommended for fresh fruit production [21,26].

The taste is better in the case of black fruits, which is due to the lower pH value compared to white fruits, which are sometimes characterized as tasteless. All these own connections, together with the nutritional and medicinal ones, make the black mulberry fruits increasingly sought after and studied [2,21,23,27,28,29,30,31].

The mineral content of mulberry fruits depends on the species, fruit maturity and composition of soil and environmental conditions (light, humidity, temperature, altitude) [21,32,33,34]. In the study of Ercisli & Orhan, 2007 [21], ten elements were determined from mulberry fruits collected from Turkey, where potassium was predominant.

Iron, an essential mineral and very rare in berry fruits, has a high value of 4.2 mg/100 g in *M. alba* and *M. nigra*. In another study [32] of the macroelements, N, K and *p* are found in large levels, while sodium is present in a very low concentration (0.01 mg) in *M. alba* and *M. nigra* grown in Spain. The levels of iron varied between 28.20 to 46.74 mg/kg and 23.92 to 37.09 mg/kg in *M. alba* and *M. nigra*, respectively, demonstrating good sources of non-heme iron. In the black mulberry, grown in Western Serbia at different altitudes, the highest amount of minerals was determined for phosphorus and potassium, with a significant difference depending on altitude. Additionally, the authors showed that black mulberries are a good source of iron, with the highest content (1.95 mg/100 g) found at 187 m altitude [33]. The form of iron in foods is ferric iron (Fe^3+^), which is less bioavailable than ferrous iron (Fe^2+^). In vivo, the increase in iron bioavailability was attributed to ascorbic acid’s chelating and reducing properties [35]. The high content of vitamin C and iron from mulberry resulted in the better bioavailability of iron and can be used to treat anemia [36].

In the mulberries fruits grown in Xinjiang region, China, potassium was the dominant macronutrient, followed by calcium and magnesium. Due to the high Fe content of Russian mulberry and black mulberry from China (11.4–11.9 mg/100 g fw), they can be used as dietary supplements to treat the iron deficiency, anemia [36].

In terms of the fatty acids profile of mulberry fruits, linoleic acid is dominant, followed by palmitic acid and oleic acid, the latter being detected only in *M. alba* and *M. nigra* [21,26,37]. Additionally, the high level of linoleic acid (52.3%) was found in white mulberry cultivated from Xinjiang region, China [36].

Regarding protein content, the mulberry fruits grown in southeastern Spain are a good source of protein; *M. alba* had a higher protein content than *M. nigra* [32]. In the mulberries cultivated in China, the ratio between essential amino acids (EAA) and total amino acid (TAA) was 44%, 42% and 29% for Russian, white and black mulberries, respectively. Foods with EAA/TAA ratio 40% are an ideal protein source, suggesting that the Russian and white mulberries cultivated in China could be used as an important quality protein source [36].

Ascorbic acid (vitamin C), a powerful water-soluble antioxidant compound, was quantified with the reflectometer in the mulberry fruits (*n* = 30); the highest value was recorded in *M. alba* (22.4 mg/100 mL), followed by *M. nigra* and *M. rubra* with 21.8 and 19.4 mg/100 mL, respectively [21]. Black mulberries had the highest content in ascorbic acid (48.4 mg/100 g fw), compared with Russian and white mulberry fruits (5.64 mg/100 g fw and 6.01 mg/100 g fw, respectively) grown in China [36].

Figure 2 summarizes the nutrients in mulberry fruits with huge importance in human metabolism.

Nguyen et al., 2008, mentioned that unripe mulberry, i.e., the green parts of fruits, contain a white sap that may be toxic, stimulating, or mildly hallucinogenic [23].

### 3.2. Morus Leaves

The leaves of mulberry are just as valuable as the fruits, not only because they are the only known source of food for the development of silkworms (*Bombyx mori*), but also because they contain, in addition to bioactive compounds, vitamins (C, B_1_, D), organic acids, minerals and proteins [16,27,38,39].

The proximate compositions of three different varieties of mulberry leaves were investigated by Iqubal, 2012 [40]. The high ash content was found in *M. rubra*, followed by *M. nigra* and *M. alba*, indicates the presence of considerable amounts of inorganic nutrients in leaves. Among the three varieties of mulberry, *M. alba* had the highest lipid content, the *M. rubra* had the highest protein level, while *M. nigra* contained high amounts of fiber. Based on their chemical composition, mulberry leaves can present dietary sources with promising nutritional values [40].

The leaves of *M. alba* and *M. nigra* possess a high iron content (119.3–241.8 mg/kg), while the other minerals are found in low concentrations. Among the macronutrients, Ca was found to be predominant, followed by N, K and Mg [26,32].

The protein content in mulberry leaves varied between mulberry species grown in Spain. The leaves of *M. alba* had a higher protein content (ranging from 14.1 ± 0.4 to 19.4 ± 0.7 % dw) than the leaves of *M. nigra* (from 13.4 ± 0.3 to 18.7 ± 0.7 % dw) [41].

Mulberry foliage is especially used to feed silkworms, but also other animals such as cattle, goats, and pigs [26,42,43]. New studies showed that *M. nigra* leaves can be introduced into pig feed in controlled concentrations as an alternative source of protein, without adversely affecting the animals. Mulberry leaves, reporting beneficial effects on the quality of the meat and the chemical composition of the muscles, the growth and finishing performance of their carcass, reduce the thickness of back fat (longissimus dorsi muscle) and increase the fat deposition in the muscles, crude protein levels and amino acids in muscle tissue [42,43].

### 3.3. Mulberry Twig and Root Bark

Mulberry twigs contains arabinosis, glucose, fructose, maltose, stachyose, tannin, and are also used in medicine, with a series of beneficial effects on serious diseases that affect the human body [4,44]. So far, no data are available on the content of minerals, carbohydrates, lipids, or root bark proteins, but bioactive compounds were identified, which are detailed in the next chapter.

## 4. Phytochemical Composition of the Botanical Parts of *Morus* sp.

Polyphenols are secondary metabolites with an important and varied role, widely found in the plant kingdom and constituting a large class of phytochemicals. Among the phytochemical compounds present in *Morus* sp., flavonoids predominate. Table 1 presents recent data from the literature (the last years) on the screening of phytochemical content found in both the fruits and leaves of *Morus* sp. Flavonoids are present in high concentrations in the epidermis of leaves and fruit skin, but also in the bark, and have a remarkable spectrum of biological activities [45,46,47,48].

### 4.1. Mulberry Fruits

The total phenols and flavonoids of *M. nigra* fruits varied between 485.5 to 1580 mg GAE/100 g fw, and 129.2 to 219.12 mg QE/100 g fw, respectively. In the case of *M. alba*, the total phenols and flavonoids varied between 60.4 to 663 mg GAE/100 g fw and 217 to 370 mg QE/100 g fw, respectively (Table 1). Anthocyanins, water-soluble pigments, are found in *M. nigra* fruits [34,60]. Many studies showed that anthocyanins are powerful antioxidants belonging to a group of flavonoids, and their color depends on the structure and presence of copigments and the acidity of the environment [9,59,61,62,63,64]. The color of black mulberries is due to the presence of anthocyanins and their content varies between 81.36 to 206.1 mg cyanidin-3-O-glucoside equivalent/100 g fw (Table 1). In generally, the contents of bioactive compounds in *M. nigra* L. are higher compared to *M. alba* L. due to the presence of anthocyanins [30,65]. Although, the presence of anthocyanins was detected in the case of some varieties of white mulberry [50,56,57], because the ripening of the fruit is accompanied by the change of color from green (unripe) to white and light purple (at full maturity) due to the accumulation of anthocyanins.

There are several studies that reported polyphenolic compounds and their functionality in mulberry fruits, and the amount varies considerably depending on the variety, as well as the climate, soil, agricultural and processing conditions [4,6,7,32,34,45,66].

The effects of ethanol concentration (20, 50 and 80%), the extraction method and time on the total yield of phenols, flavonoids and anthocyanins from black, red and white mulberry from southeast Serbia were investigated [67]. The content of total phenols of *M. nigra* depends on the extraction method and ranges from 60.04 to 150.13 mg GAE/100 g fw and 69.11 to 142.18 mg GAE/100 g fw for maceration and ultrasonic extractions, respectively. In terms of flavonoid content, *M. rubra* and *M. white* have a lower content of flavonoids than *M. nigra*. The content of the monomeric anthocyanins ranges from 64.08 to 137.06 mg cyanidin-3-O-glucoside/100 g fw for ultrasonic extraction. Ultrasonic extraction was the most efficient in all samples due to the intensification of mass transfer, and thus the penetration of the solvent into the mulberry cells was easier [67].

In recent years, due to the development of analytical techniques for the separation and identification of compounds, it was possible to identify the composition of mulberry metabolites [8,44,68,69,70,71,72].

In *M. nigra*, there are four main anthocyanins, where cyanidin-3-O-glucoside is the main pigment (Appendix A) [38,65,73,74].

Thirty-five metabolites were identified in white and black mulberry fruits from Italy (Campania region) [72]. Four anthocyanins from *M. nigra* were identified, these being pelargonidin and cyanidin conjugated with one, two or three hexose sugars (Appendix A). The flavonol compounds, quercetin and kampferol, esterified with one to three sugars or one or two sugars with one malonyl group, were identified in both white and black mulberries (Table 2). Two flavanonols were detected in black mulberries, dihydroquercetin (taxifolin) hexoside and dihydrokaempferol hexoside, which are precursors of the biosynthesis of anthocyanidins, such as cyaniding and pelargonidin, respectively [72,75].

The total anthocyanin contents of 12 mulberry cultivars in Korea depended on the variety and ranged from 0.51 to 28.61 mg/g dw. Cyanidin-3-O-glucoside was the major anthocyanin compound, followed by cyanidin-3-O-rutinoside, and the lowest content was recorded in the case of pelargonidin-3-O-glucoside [56]. Using UHPLC-(ESI)-qTOF, 16 anthocyanin compounds were identified, including six that were identified for the first time. These are malvidin hexoside, cyanidin malonyl hexose hexoside, cyaniding pentoside, cyanidin malonyl hexoside, petunidindeoxyhexose hexoside, and cyanidin deoxyhexoside [56].

A complex study investigated and compared different medicinal parts of *M. alba* (root barks, twigs, fruits, leaves) in terms of chemical composition [59]. In the white mulberries, chlorogenic acid, rutin, isoquercitrin were identified, but in lower levels than in the leaves.

In *M. nigra* fruits, the main flavonoids identified were quercetin 3-O-glucoside and rutin (quercetin 3-rutinoside) (Appendix A) [38]. Another study showed that the main compounds present in mulberry fruit come in the various glycosylated forms of quercetin and kaempferol, and the content of quercetin glycoside was found to be higher than the amount of kaempferol-O-rutinoside [32].

The major phenolic group profile found in *M. nigra*, according to the amount recorded, was phenolic acids (benzoic acid and cinnamic acid derivatives), followed by flavonols and anthocyanins. Protocatechuic acid is the main derivative of hydroxybenzoic acid, while chlorogenic acid is the predominant hydroxycinnamic acid. In other studies, a different pattern was found, where anthocyanins were higher than the amount of flavonols [2,79,80,81].

Miljkovic et al., 2015, identified the presence of anthocyanins in black and red mulberries, other than those mentioned above: cyanidin-3-O-arabinoside, peonidin-3-O-arabinoside, delfinidin-3-O-arabinoside, peonidin-3-O-glucoside or galactoside or malvidin-3-O-arabinoside, delfinidin-3-O-glucoside or galactoside, petunidin-3-O-glucoside or galactoside, malvidin-3-O-glucoside or galactoside [82].

Özgen et al., 2009, investigated the phytochemical profile and antioxidant properties of the anthocyanin-rich mulberry species, *M. nigra* and *M. rubra*, harvested from Turkey. Their study showed that *M. nigra* had the richest amount of anthocyanin, with an average of 571 μg cyanidin-3-glucoside equivalent/g FW (fresh weight) compared to *M. rubra* [83].

### 4.2. Mulberry Leaves

Flavonols are the main group of flavonoids in mulberry leaves, whose concentrations differ depending on the geographical area. In Spanish mulberry leaves, flavonols range from 3.7 to 9.8 mg/g dw [32], the Polish variety of *M. alba* leaves contain 17.6 mg/g dw [84], while Tunisian *M. rubra* leaves reach up to 85.45 mg/g dw [77].

In terms of caffeoylquinic acid derivatives, neochlorogenic acid, chlorogenic acid, cryptochlorogenic acid, caffeoylquinic acid isomer and caffeoylquinic acid glucoside were identified in the leaves of *Morus* [32]. In general, chlorogenic acid is predominant in mulberry leaves (Appendix A), for example in *M. alba* leaves reaching 51.04% from total phenolic acids [84]. In contrast, syringic acid was the major phenolic acid identified in *M. nigra* leaves, and is considered the key compound for the biological effects of mulberry leaves [71,85,86].

The phytochemical composition of mulberry leaves belonging to eight clones from Spain (white and black) was investigated [81]. The flavonol derivatives, mainly quercetin and kaempferol in glycosylated forms were identified in mulberry leaves. Among the quercetin derivatives detected in *Morus* leaves, the compounds quercetin malonyl-dihexoside, quercetin-rhamnose-hexose-rhamnose and quercetin-malonyl-rutinoside were identified for the first time. Concerning kaempferol derivatives, kaempferol rutinoside hexoside, kaempferol-malonyl-dihexoside and kaempferol-malonyl-rutinoside were identified for the first time.

In the leaf samples of *M. alba*, rutin, isoquercitrin, astragalin, skimmin, and chlorogenic acid were predominant [87].

In the leaves of mulberry, it was also found that the predominant glycosides of flavonol were rutin and quercetin 3 (6-malonylglucoside), which are flavonol glycosides mainly responsible for the antioxidant capacity of the leaves, and have properties that reduce oxidative stress in the liver and improve hyperglycemia [88,89,90,91].

Other flavonoids were identified in mulberry leaves, such as mornigrol E, mornigrol F and morusin, the latter being a prenylated phenolic present in all parts of the mulberry plant, the largest amount being in the bark of the root, while its content decreased in the following order: stem, branches and leaves [40,63].

### 4.3. Mulberry Twig and Root Bark

The root barks of mulberry, especially of *M. alba* L., were widely used in Chinese herbal medicine for treating various diseases. The main constituents of root barks are prenylated flavonoids such as morusin, kuwanon C (known as mulberrin) and kuwanon G [92,93]. Another kind of polyphenols found in mulberry are Diels–Alder-type adducts, most of which contain flavonoid groups, and the C-2 and C-3 of the flavonoid unit could be replaced by prenyls and their correspondents [94].

Morusalbanol A, a neuro-protective DielseAlder adduct was isolated from the bark of *M. alba* grown in China [78]. Ha et al., 2018, investigated phytochemicals from the root bark of *M. alba* collected from Ulsan province, Korea, and identified four Diels–Alder-type adducts (morusalbins A-D), one isoprenylated flavonoid (albanin T), among which 21 other known phenolic compounds were found [95]. A phytochemical investigation of the root bark of *M. alba* (Korea), and their anti-Alzheimer’s disease activity, was evaluated by Kuk et al., 2017 [96]. The authors identified two Diels–Alder-type adducts (mulberrofuran G and albanol B) and one prenylated flavonoid (kuwanon G).

Morusin, a prenylated flavonoid, from the perspective of chemical structure has hydroxy groups at 5 (from A ring), 2′ and 4′ (from B ring), a prenyl unit at position 3, and dimethyl pyran group across positions 7 and 8 (Appendix A) [88].

The major flavonoid constituents, isolated from root barks of *M. alba* and collected in Korea, were kuwanon E, (3′-prenylated flavanones), kuwanon G, (tetrahydroxyflavone), and norartocarpanone, a member of flavanones [92].

The position and number of hydroxy and prenyl groups on the flavone backbone influences their bioactivities [94,97].

The phenolic phytochemicals in mulberry bark were found to be rich in oxyresveratrol, resveratrol *p*-coumaric acid, chrysin, catechin, vanillic acid, ferulic acid, chlorogenic acid, mulberroside A, maclurin, and moracins [98,99].

The stilbene, mulberroside A and prenylated flavonoids, kuwanon G, and morusin identified in the root barks were higher than in twigs. Instead, oxyresveratrol content in twigs was higher than in root barks [87]. Oxyresveratrol (2,3′,4,5′-tetrahydroxystilbene) is a natural stilbene, a phytoalexin that has recently received attention due to its therapeutic potentials [100].

Mulberry root bark also contains other flavonoids, such as mulberrin, mulberrochromene and cyclomulberrin [4,76,101].

## 5. Health Benefits and Effects

The presence of polyphenolic compounds of the genus *Morus*, have recently received increased attention and interest due to its biological effects, such as anticancer, anti-inflammatory, antidiabetic, antihypertensive, antinociceptive, antiaging, antianemic, antibacterial and antioxidant activities [26,102,103,104,105].

*M. nigra* is recognized as an appreciable source of flavonoids, especially anthocyanins, which are considered to possess certain protective effects on human health. *M. nigra* fruits are of huge importance for medicine and pharmaceuticals production, being recommended for the prevention and treatment of serious diseases such as diabetes, cancer, bacterial infections, arthrosclerosis, hyperlipidemia, neurological diseases, inflammation, hypertension, or to improve drugs with their natural flavor and color [1,21,23,27,45,102]. In addition, due to the anthocyanin pigments responsible for the black/purple color, mulberry fruits are also used as color or flavor additives in the food industry [104].

### 5.1. Antioxidant Potential

Oxidative stress is an imbalance between free radicals and antioxidants in favor of radicals. The high level of free radicals induces many biochemical changes in organisms, with an important contributing factor to a vast number of diseases such as diabetes, atherosclerosis, cardiovascular and neurodegenerative diseases, and cancer [4,6,84,106,107]. Some free radicals are naturally produced in the body and others come from different sources, mainly from polluted environments [108]. Free radicals can alter important classes of biological molecules, thus altering the body’s normal redox state, resulting in increased oxidative stress [109,110,111]. Increasing the level of exogenous antioxidants in the body after their intake can help maintain a good antioxidant status. Antioxidants can counteract the effects of oxidative damage caused by free radicals by preventing and repairing damage caused by reactive oxygen species and reactive nitrogen species; therefore, they can increase immune defense and reduce the risk of cancer, cardiovascular disease, cataracts, diabetes and other degenerative diseases [55,63,72,110,112,113]. A variety of methods were developed to evaluate in vitro the antioxidant capacity of food components [109]. The most commonly used methods are based on the single electron transfer (SET) mechanism, including the Folin–Ciocalteu assay, ABTS, DPPH, FRAP etc. [109]. The methods are based on the ability of samples to scavenge a synthetic-colored radical or to reduce the redox-active compound, which is spectrophotometrically monitored. Assessing the antioxidant capacity of a sample by a single method is not relevant; therefore, several procedures should be carried out to investigate the antioxidant capacity of foods [109].

The in vitro antioxidant capacity of fruits and leaves from different mulberries species, evaluated through various methods, are presented in Table 3.

The main compounds that contribute to the antioxidant capacity, identified in *M. nigra* fruits, are phenolic acids, flavonoids and anthocyanins that promote beneficial effects on human health [26,117].

A comparative study between black and white mulberry methanol and acetone extracts revealed that *M. nigra* exhibited the highest antioxidant capacity compared with *M. alba* using the ABTS, DPPH and reducing power assays [118].

Sánchez-Salcedo et al. (2015) determined the antioxidant activity of white and black mulberry fruits, and results showed a significant difference between the clones from 3.84 to 20.73 mg Trolox/g dw and 3.62 to 12.91 mg Trolox/g dw for ABTS and DPPH methods, respectively. In general, the antioxidant activity was significantly higher in *M. nigra* fruits than in *M. alba* fruits, and an intra-species variability was observed. The MN1 black mulberry clone showed a significant antioxidant potential (*p* < 0.05) higher than the rest of the clones [32].

Other studies showed that *M. nigra* has a high antioxidant activity in the method of bleaching β-carotene (>70% inhibition). In addition, the anthocyanin mulberries (cyaniding derivates) showed the highest liposome antioxidant activity [119].

Mulberry leaves also have a strong antioxidant capacity, and the predominant antioxidants are kaempherol and quercetin, which effectively protect red blood cells against oxidative damage caused by free radicals [5,40,103].

The methanol extract of *M. nigra* leaves enhanced antioxidant enzymes such as SOD, GSH-Px, CAT in serum, brain and liver tissues, and reduced MDA level, effects that are related to the improved cognitive impairment in mice D-galactose group. Vanilic and chlorogenic acids are the major compounds in the extract and are responsible for antioxidant actions [120,121].

Different parts of the *Morus* species—the leaves, root and fruits of *M. nigra*—have a high content of antioxidants. Morusin is a powerful antioxidant of the root bark and was reported to be a promising candidate in the treatment of cancer, including lung cancer, but also attracted scientific interest for its strong antinociceptive and antitumor properties [88,101,117].

Mulberry leaves and fruits contain resveratrol and oxyiresverrol, both of which are natural dietary phenolic compounds, considered powerful antioxidants. *M. rubra* fruits contain the highest content of resveratrol (50.61 μg/g dw) compared to jamun fruits (*Syzygium cumini* L.) and jackfruit. The scavenging effect of the DPPH radical is dependent on the concentration dose, and IC50 was found to be 0.40 mg/mL [122]. Resveratrol was studied for its cardioprotective, neuroprotective and antiviral properties. Due to the fact that its antiviral activity was demonstrated in the fight against several viruses, it was recently considered to be an anti-inflammatory agent against coronavirus, a severe acute respiratory syndrome type 2 [123,124].

Mulberry leaves are recognized for their beneficial effects on health, and various studies have shown that they protect red blood cells against oxidative damage, due to their content in flavonoids, such as quercetin and kaempferol [31,103].

In a study performed on rats, the phytochemical profile of *M. nigra* leaves and their hypolipidemic effect were evaluated. The results of this study demonstrated the potential therapeutic effects of *M. nigra* leaf infusion, due to large amounts of phenols, flavonoids and antioxidant activity. The high content of polyphenols, mainly chlorogenic acid, normalized the hyperlipidemic disturbance of rats. The antioxidant capacity of mulberry leaves depends on the extraction methods, the highest DPPH-scavenging activity was recorded in the infusion and hydrometanolic extracts (83.85± 0.99% and 81.71 ± 0.05%, respectively) [103].

The antioxidant capacity of *M. nigra* leaves was evaluated by DPPH and β-carotene/linoleic acid co-oxidation system assays depending on the seasonal variations. A high antioxidant level highlighted that black mulberry leaves were rich in phenolic compounds (syringic acid, quercetin and rutin) that strongly contributed to antioxidant capacity [71].

Morusin, a prenylated flavonoid identified in the root bark of *Morus* is a powerful antioxidant agent due to its chemical features, reducing ROS formation and inflammation [88].

Interest in natural antioxidants from mulberries has grown considerably in recent years and new approaches are being explored for their most effective extraction. [112,125,126]. Different extraction techniques and modern separation technologies were used to separate and isolate the bioactive components from mulberry [125].

### 5.2. Anticancer Activity

The International Agency for Research on Cancer (IARC), a specialized agency of the World Health Organization (WHO), reported that by 2020, about 10 million deaths worldwide were caused by cancer, with Europe ranking second (19.6%) after Asia (58.3%). Due to drug resistance and side effects, there is an urgent need to study and develop new therapies for cancer treatment [127].

The anticarcinogenic mechanisms of mulberry phenols demonstrated in several cell cancer lines or in animal tumor models include antioxidant activity, antiproliferation, the induction of apoptosis and antiangiogenic activity [2,128,129,130].

The antioxidant compounds in *M. nigra* fruit, leaves and bark, were shown to have an anticancer effect in addition to other properties [47,48,131]. The bioactive components in *M. nigra* leaves were proven to be a good candidate in the genetic protection and treatment of organotoxicity induced by oxidative stress. The methanolic extract of mulberry leaves had an effective cytotoxic behavior against cancer cells [1,132].

The anthocyanins found in *M. nigra* fruits, such as cyanidin and pelarginidine, had the ability to reduce the viability of cancer cells and inhibit tumor progression. Hydroxynamic acid derivatives of mulberry fruit were shown to increase reactive oxygen species production by acting as prooxidants, and thus destroying cancer cells [2,133,134].

Cho et al., 2017, isolated cyanidin-3-glucoside from mulberry fruit, showing that it had the ability to inhibit tumor growth in nude female mice inoculated with MDA-MB-453 cells, showing active apoptosis by caspase-3 cleavage and DNA fragmentation by proteins of the Bcl-2 family and Bax [133].

Shu et al., 2021, isolated guangsangon E compound from white mulberry leaves, with the ability to inhibit cell proliferation, induce autophagy and apoptosis in lung and nasopharyngeal cancer cells, and increase the flow of autophagy by increasing lysosomal activity and the fusion of autophagosomes and lysosomes. Guangsangon E activates the stress of the endoplasmic reticulum, which is involved in inducing autophagy, by accumulating reactive oxygen species, eventually attenuating autophagy-mediated cell death [135].

Mulberry bark has, in addition to anti-inflammatory and antioxidant activities, antitumor and anticancer effects due to the presence of morusin, which was reported to be a promising candidate for the treatment of certain types of cancer, including lung cancer [45,88,91,136].

The potential of morusin to inhibit the growth of breast cancer cells was confirmed by some in vitro and in vivo studies through C/EBPβ and PPARγ-mediated lipoapoptosis, and induced the adipogenic differentiation, apoptosis and lipoapoptosis of cancer cells [137]. Morusin inhibited the growth and migration of human cervical cancer stem cells by decreasing the expression levels of NF-κBp65 and Bcl-2 and Bax, and increased caspase-3 in a dose-dependent manner [138]. Morusin has a strong antitumor activity for human hepatocellular carcinoma in vitro and in vivo, by inducing apoptosis and inhibiting angiogenesis; moreover, it inhibited the proliferation, migration and tube formation of human umbilical vein endothelial cells [139].

### 5.3. Antidiabetic Therapy

Diabetes is associated with increased oxidative stress. Hyperglycemia, excess fatty acids and insulin resistance lead to an increase in oxidative stress, followed by other metabolic disorders such as obesity, diabetes, high blood pressure and cardiovascular disease [113,140].

Many research studies reported the phytochemical and pharmacological aspects of *M. nigra* as an antidiabetic drug [103,113,141] and found that all botanical parts (leaves, root, branches and fruits) have special effects on antidiabetic activity [26,72,113,142].

Leaf and leaf tea is considered a healthy and suitable food in the fight against diabetes, regulating energy metabolism, improving the blood lipid profile, improving insulin sensitivity and lowering blood glucose [143,144]. The main compounds in mulberry leaf tea that were reported to have an antidiabetic effect are phenolic acids, flavonoids, polysaccharides, and 1-deoxynojirimycin, the latter has the effect of lowering high levels of sugar blood [103,145].

Some components identified in mulberry leaves, such as tricetin, gallic and chlorogenic acid, possess the ability to regulate the insulin receptor substrate 1, glycogen synthase kinase-3 beta, interleukin 6, and other proteins responsible for insulin signaling pathway, glucose metabolism, nephropathy diabetes, obesity and other diabetes-dependent factors [146].

In traditional Egyptian culture, black mulberry leaves were also used as a remedy for the treatment of diabetes, improving the redox profile and glycemic response in the liver [113,147]. In an in vivo study of streptozotocin-induced diabetic male albino rats by Hago et al., 2021, the leaf extract caused a significant decrease in blood glucose. In vitro model confirms the anti-diabetic effect of mulberry leaf extract, which has the ability to reverse hyperglycemia, strongly inhibit the α-glucosidase enzyme and prevent the liver and kidney damage associated with diabetes [147].

*M. nigra* fruits and seeds are also used to treat diabetes, especially type II diabetes and its associated complications [4,140].

*M. alba* fruit polysaccharides repaired damaged pancreatic tissue, significantly lowered fasting blood glucose levels, curved the oral glucose tolerance zone, and significantly increased high-density lipoprotein cholesterol and total cholesterol of a high-fat diet in streptozotocin-induced type 2 diabetes in rats. Additionally, in the same model, there were significant decreases in fasting serum insulin levels, the homeostasis model for assessing insulin resistance, serum alanine transaminase levels and triglyceride levels [148].

Mulberry twigs and bark are also known to contain polyphenols and polysaccharides. Thus, *M. nigra* twig methanol extract is a tyrosinase inhibitor and has the ability to lower blood sugar [105,148].

Ha et al., 2020, isolated the farnesylated 2-arylbenzofuran compound from *M. alba* extract which exerts an anti-obesity effect and is a potent inhibitor of the intracellular protein tyrosine phosphatase 1B (PTP1B), which plays a critical role in the insulin receptor signaling pathway [149].

### 5.4. Immunomodulatory Effect

Immune stimulation in defense against disease in humans is currently receiving much attention, and there is a growing interest in the discovery of natural products with the effect of strengthening immunity. The immunomodulatory effect of plants has gained great interest, especially in recent years due to a human awareness to modulate the immune system [150].

The bioactive compounds present in the botanical parts of the *M. nigra* plant have immunomodulatory properties on infectious diseases of bacterial, parasitic and viral etiology. Thus, the mulberry could be exploited to find new avenues for the development of food supplements with a strong immunostimulatory and immunosuppressive effect to enhance quality of life [132,151].

For example, resveratrol is one of the compounds present in *M. nigra* as a powerful antioxidant with antiviral activity, and recent studies suggested that it has a therapeutic effect in homeostatic disorders associated with COVID-19 and is a potential adjunct to COVID treatment [124,152,153,154].

*Mulberry* root bark has been used for medicinal purposes since ancient times [155,156,157]. It has an acidic taste, as well as anthelmintic and cathartic, anti-inflammatory, vermifuge and vermicide, diuretic, anti-inflammatory, antitussive and antipyretic properties in oriental medicine, but is also used to treat high blood pressure [42,45,102,158]. The juice extracted from the mulberry root fluidified the blood, and it is very useful for fighting tapeworms and other parasites in the digestive tract [159,160,161].

The immunomodulatory effect generated by both the leaves and fruits of *M. alba* was proven. Orally administered methanol *M. alba* leaves extract increased serum immunoglobulin levels and prevented mortality induced by bovine *Pasteurella multocida* in Swiss albino mice. Additionally, it significantly increased the phagocytic index in the carbon clearance test, and the circulating antibody titer in the indirect haemagulation test was recorded as having positive effects on both humoral and cell-mediated immunity [132].

Recently, it was reported that *M. alba* fruits induce the phenotypic maturation of dendritic cells due to the polysaccharides in the composition, and can thus be used as an adjunct in dendritic-cell cancer immunotherapy [128].

Numerous studies confirmed that various parts of mulberry have various benefits for human health, such as anti-inflammatory, neuroprotective, antiatherosclerosis, hepato- and gastro-protective, antibacterial, anti-melanogenesis properties, which are presented in Table 4.

## 6. Applications of Mulberry in the Food Industry

Epidemiological studies suggest the role of oxidative stress in the generation and propagation of many chronic diseases. Therefore, to counteract the unwanted effects of oxidative stress exogenous antioxidants in the form of dietary supplements or even functional foods are necessary for the human body [40].

Black mulberry fruits are considered functional foods that, when ripe, have a black–purple color and can be eaten fresh or dehydrated [165,172,173]. Baked mulberries are very perishable fruits, mainly due to their smooth texture, high softening and breathing rate, and susceptibility to fungal attacks. After harvesting, their aroma and appearance change, decomposition processes increase, and synthesis processes are reduced in intensity, which is why an optimal method of preservation is recommended [172,174]. In order to satisfy the requirements of consumers by ensuring that products are both healthy and delicious, and since mulberries are famous all over the world, they have started to be processed in different forms, so that they can be stored and consumed in the long term [172,175,176]. Due to the anthocyanin pigments responsible for their dark color, mulberry fruits are also used as color or flavor additives in the food industry for the production of healthy foods without synthetic food additives [165,176].

In recent years, the *Morus* plant species began to occupy an important position in the food industry, due to its health benefits (Table 4). Recently, various mulberry-based foods were developed and marketed in Asian regions [41]. As a food, in addition to being consumed fresh, but also dried, mulberry is suitable for the preparation of several foods, such as juices, syrups, wine, vinegar, brandy, jellies, marmalades or jams [27,42].

Mulberry fruits can be eaten ripe or can be found in the market in various forms as nutritional supplements, as a tonic and sedative. The only official medicinal product in the British Pharmacopoeia is *Siropus mori*, used mainly as an adjunct to its slightly laxative and expectorant qualities [5,102,158].

The water-soluble anthocyanin pigments responsible for the color of mulberry fruits are also used as color or flavoring additives in the food industry, a practice increasingly used to create foods that contain natural additives [165,176].

Taking into account the requirements of consumers, the nutritional value and bioactive compounds that provide a number of beneficial effects on the human body, the mulberry plant is becoming increasingly studied or even exploited for the production of food or food supplements. It is also used as a main ingredient in the production of foods, such as jams [177], lollipops and jellies [178], wines [179,180], syrups [181] and other functional beverages and foods [145,182,183,184,185,186] (Table 5).

The mulberry plant is also used in the form of food supplements. In recent years, a number of patents (Table 6) were developed based on mulberry with various functional and therapeutic applications.

Mulberry fruits possess a lower pectin content than other fruits used in the manufacture of jams. There is a patent (RO 135033 A2) that implements different processes for obtaining jams from black, white or red mulberry fruit by gelling with the addition of pectin [189]. An ethnobotanical study conducted by Li et al., 2017, on white mulberry leaf tea, showed that teas could be included in the category of functional foods, having a detoxifying effect, treating coughs and sore throats, colds, etc. [190]. Lin et al., 2020 [187], confirmed that drinks obtained from mulberry leaves are functional products that prevent diseases related to aging and Meng et al., 2020, mention that white mulberry leaf tea is used in Asian countries in order to control diabetes [104].

**Table 6 plants-11-00152-t006:** List of patents based on the therapeutic and functional applications of *Morus* (2017–2021).

Application No.	Species/Part	Sample Type	Results/Mechanism	Ref.
US 11,090,349 B2	*Morus alba* L.; *Morus alba var. multicaulis* L.; *Morus nigra*; *Morus australis Poir.*	Raw material, dry leaves	Inhibits α-glucosidase. It has the ability to control blood glucose levels and reduce melanin production for the treatment of conditions caused by pigmentation, such as freckles, chloasma, striae gravidarum, sensitive plaque and melanoma.	[191]
AU 2019201188 B2	*M. alba* root bark; acacia barks; *Uncaria gambir*, leaves; *Curcuma longa* L.	Mixture extract	The compound mixture, demonstrated beneficial synergistic effects with improved anti-inflammatory and anti-nociceptive efficacy, but also the attenuation of joint stiffness.	[192]
US 10,588,927 B2	Mulberry (*M. alba*) and poria cocos peel	Mixed extract	Used either as a food product or as a pharmaceutical composition with the aim of preventing or treating degenerative neurological diseases, having the ability to improve memory and protection on neurons.	[97]
US 2020/0360457 A1	*M. alba* and *M. nigra* root	Macerate extract	As an active ingredient, at least one extract from the root of the plant is used, according to the invention. It is rich in moracenine A, moracenin B, kuwanon C, wittiorumin F and mulberrofuran T, also used in cosmetic composition and a pharmaceutical or nutraceutical composition.	[193]
US 2020/0178585 A1	*Morus* sp. fruits	Savory concentrate/seasoning, with vegetable fat.	Used as a cooking aid in the preparation of starch-rich food.	[194]
US 2020/0197429 A1	Astragalus root; phlorizin; *M. alba* root bark; olive leaf and bitter melon.	Standardized extracts	Dietary supplement with the aim of controlling postprandial blood sugar.	[181]

A patented formula containing *M. alba* leaf extract (200 mg), along with berberine and red yeast rice powder has benefits on cardiovascular prevention in patients with dyslipidemia and reduces insulin resistance by protecting subjects from the side effects of smoking [195].

Yimam et al., 2017, studied the effect of UP1306, a composition based on a patented mixture of heartwood of *Acacia catechu* and the root bark of *Morus alba*, with beneficial effect on joints, evidenced by the attenuation of symptoms associated with osteoarthritis in monosodium–iodoacetate-induced rats, reducing pain sensitivity, significantly improving the integrity of the articular cartilage matrix and causing minimal subchondral bone damage [196].

An aqueous and ethanolic extract of *M. alba* was developed (US7815949 B2) with estrogenic effects, and used for the treatment of climatic symptoms, osteoporosis and breast or uterine cancer [197]. Another product (US2010/0166898 A1) from the bark of *Morus australis* Poir, containing kiwanon H, was used as an antimicrobial agent [198]. An ethanolic extract of white mulberry branches, which has a whitening effect, is the basis of another patent (US2006/0216253 A1) [199].

## 7. Conclusions

All botanical parts of mulberry trees (fruits, leaves, twigs, root bark) are rich in nutrients and phytochemical compounds, exhibiting various pharmacological properties which may help to treat various chronic diseases. This review aimed to highlight the huge potential of the *Morus* plant and its application in the food, pharmaceutical and cosmetic industries. Among the phytochemical compounds present in *Morus* sp., flavonoids are the predominant compounds. In *M. nigra* fruits, anthocyanins are the main compounds, along with the flavonols, quercetin and kaempferol, present in various glycosylated forms. Flavonols are the main group of flavonoids in mulberry leaves, and among the phenolic acids, chlorogenic acid was predominant. Mulberry root bark is an important source of bioactive compounds, in which prenylated flavonoids such as morusin, mulberrin and kuwanon G, have various health-promoting effects. According to the bioactive phytochemical composition, *Morus* species display a variety of biomedical applications including antioxidants, anticancer, antidiabetic, anti-inflammatory properties, etc. Based on these findings, white and black mulberries can be used as a promising and rich source of phytochemicals, in order to develop food supplements or functional foods to prevent or ameliorate various chronic diseases. Since mulberries are perishable, various mulberry-based products were developed and patented in the food industry as functional foods. Further studies are needed to develop fruit preservation strategies, such as the development and application of edible films or coatings to increase the safety, quality and shelf life of mulberry fruits.

## Figures and Tables

**Figure 1 plants-11-00152-f001:**
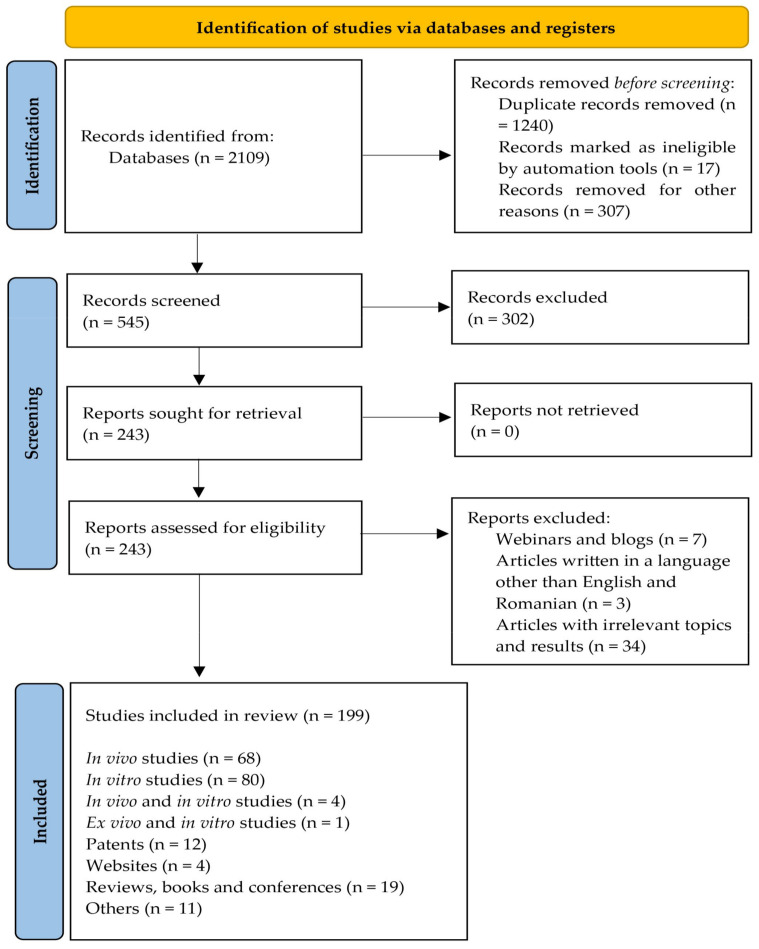
PRISMA 2020 flow diagram for the present review.

**Figure 2 plants-11-00152-f002:**
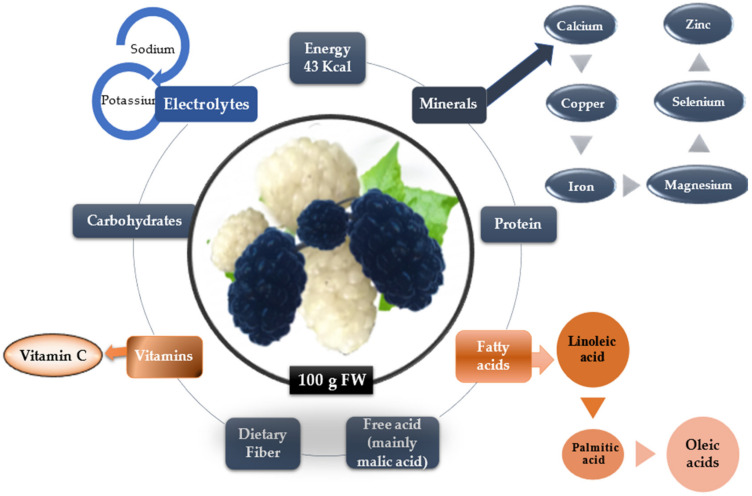
The nutrients components of mulberry fruits.

**Table 1 plants-11-00152-t001:** The content of phytochemicals (polyphenols, anthocyanins and flavonoids) from mulberry fruits and leaves from data of literature of the last years (2016–2021).

Species	Organs	Total Polyphenols	Anthocyanins	Flavonoids	References
*Morus nigra*	fruits	6.93 ± 0.58 mg GAE/g dw	233.77 ± 24.02 µg/g dw	Not recorded	[49]
		485.5 ± 7.1 mg GAE/100 g fw	206.1 ± 1.8 mg C3GE/100 g fw	Not recorded	[50]
		Not recorded	Not recorded	20.8703 ± 0.9091 mg RE/g dw	[51]
		502.43 ± 5.10 mg GAE/100 g fw	81.36 ± 2.05 mg C3GE/100 g fw	219.12 ± 4.45 mg QE/100 g fw	[8]
		1422 mg GAE/100 g dw	Not recorded	276 mg QE/100 g dw	[26]
		3.7 mg GAE/g fw	11.3–20.3 mg C3GE/g fw	Not recorded	[52]
		6585 ± 146 mg GAE/kg fw	Not recorded	1292 ± 52.7 mg QE/kg fw	[53]
	leaf	Not recorded	Not recorded	68.32 mg RE/g dw	[54]
		24.37 ± 2.14 GAE/100 g dw	Not recorded	Not recorded	[40]
*Morus atropurpurea Roxb.*	fruits	11.33 ± 0.7 mg GAE/g fw	1177.36 ± 136.14 mg C3GE/100 g fw	15.1 ± 1.22 mg RE/g fw	[55]
*Morus alba*	fruits	60.4 ± 3.1 mg GAE/100 g fw	289.2 ± 0.9 mg C3GE/100 g fw	Not recorded	[50]
		181 mg GAE/100 g dw	Not recorded	0.0816 mg QE/g dw	[26]
		663 ± 28.5 mg GAE/kg fw	Not recorded	217 ± 18.2 mg QE/kg fw	[53]
		5.68 to 40.46 mg GAE/g dw	0.51 to 28.61 mg/g dw	0.65 to 3.70 mg QE/g dw	[56]
		534.2 ± 12.46 mg GAE/g dw	15.5 ± 2.27 mg C3GE/g dw	427.6 ± 15.94 mg CE/g dw	[57]
	leaf	2.468 ± 0.05 mg GAE/g	Not recorded	Not recorded	[58]
		51.43 ± 1.11 mg GAE/g dw	Not recorded	43.75 ± 0.78 mg QE/g dw	[59]
		16.21 ± 1.34 mg GAE/g dw	Not recorded	26.41 ± 1.14 mg RE/g	[40]
*Morus rubra*	fruits	1035 mg GAE/100 g dw	Not recorded	219 QE/100 g dw	[26]
	leaf	Not recorded	Not recorded	31.28 ± 2.12 mg RE/g	[40]

GAE—gallic acid equivalent, C3GE—cyanidin-3-glucoside equivalent, RE—rutin equivalents, QE—quercetin equivalent, CE—catechin equivalents, fw—fresh weight, dw—dry weight.

**Table 2 plants-11-00152-t002:** Summary of phenolic components isolated from different parts of mulberry tree across-studies (2017–2021).

Species	Organs	Type of Sample	Technique	Identified Components in *Morus* sp.	References
*M. nigra*	fruits	Lyophilized samples	HPLC-DAD-ESI HRMS	Anthocyanins: cyanidin-hexoside, cyanidin-pentosyl-hexoside, cyanidin-rhamnosyl-hexoside, cyanidin-sambubiosyl-rhamnoside, cyanidin-sambubiosyl-glucoside; delphinidin-pentoside, delphinidin-dirhamnosyl-hexoside, petunidin-pentoside, peonidin-hexoside Proanthocyanidin (condensed tannins): procyanidin trimer 1 Flavonols: kaempferol-rhamnoside, kaempferol-hexoside, kaempferol-malonyl-hexoside, kaempferol- rhamnosyl-hexoside, kaempferol-dihexoside; quercetin-rhamnoside, quercetin-hexoside, quercetin-malonyl-hexoside, quercetin-rhamnosyl-hexoside, quercetin-dirhamnosyl-hexoside, quercetin-rhamnosyl-dihexoside, rutin; myricetin-hexoside Flavone: apigenin-hexoside, apigenin-dihexoside Phenolic acid: chlorogenic acid	[68]
		Freeze-dried samples	LC-MS	Anthocyanins: cyanidin hexoside, cyanidin hexose-deoxyhexose; pelargonidin hexoside, pelargonidin hexose-deoxyhexose Flavonols: quercetin-3-O-rutinoside, quercetin-3-O-rutinoside-7-O-glucoside, quercetin-malonylhexoside, quercetin-hexoside, quercetin hexoside malonyl hexoside, quercetin hexose hexose, quercetin-hexose-hexose-deoxyhexose, kaempferol-3-O-rutinoside, kaempferol-3-O-rutinoside-7-O-glucoside, kaempferol-hexoside, kaempferol hexoside malonyl hexoside, kaempferol hexose-hexose deoxyhexose, kaempferol malonyl hexoside Flavononol: taxifolin hexoside; dihydrokaempferol-hexoside.	[72]
		Mulberry dry powder	UPLC-TUV/QDa	Anthocyanins: cyanidin-3-O-glucoside, cyanidin-3-O-rutinoside, pelargonidin-3-O-glucoside Flavonols: rutin, isoquercetin, morin, quercetin, kaempferol	[76]
	leaves	Ethanolic extract	HPLC-DAD	Anthocyanins: cyanidin; Flavonols: quercetin; kaempferol. Flavonols: catechin. Phenolic acid: caffeic acid; coumaric acid.	[70]
		Aqueous extract	HPLC-PDA	Flavonols: quercetin, rutin Phenolic acid: syringic acid	[71]
	twigs	Powder	HPLC; LC-MS-MS; UV-spectra; IR-spectra	Prenylated flavonoids: morunigrols A, B, C, D; cudraflavone B; morusin; moracin C and P. Diels−Alder adducts: morunigrines A and B;	[44]
	roots bark	Air-dried roots bark.	RP-MPLC; MS	Prenylated flavonoids: kuwanon L, G and H; cudraflavanonă A; morusin; chalcomoracin, norartocarpetin. Stilbenes: oxyresveratrol	[69]
*Morus alba*	fruits	Powder samples	HPLC-DAD	Flavonols: rutin; isoquercitrin; Flavanonol: taxifolin Prenylated flavonoids: morusin; Phenolic acid: chlorogenic acid; 4-hydroxycinnamic acid	[77]
		Freeze-dried sample	LC-MS	Flavonols: quercetin-3-O-rutinoside, quercetin-3-O-rutinoside-7-O-glucoside, quercetin-malonylhexoside, quercetin-hexoside, quercetin hexoside malonyl hexoside, quercetin hexose hexose, quercetin-hexose-hexose-deoxyhexose, kaempferol-3-O-rutinoside, kaempferol-3-O-rutinoside-7-O-glucoside, kaempferol-hexoside, kaempferol hexoside malonyl hexoside, kaempferol hexose-hexose deoxyhexose, kaempferol malonyl hexoside Flavanonol: taxifolin hexoside; dihydrokaempferol-hexoside.	[72]
		Dry powder	UPLC-TUV/QDa	Flavonols: rutoside, morin, isoquercetin, quercetin, kaempferol	[76]
	leaves	Powder sample	HPLC-DAD	Flavonols: isoquercitrin; rutin; quercitrin, astragalin (kaempferol-3-O-glucoside). Coumarin: skimmin Phenolic acid: chlorogenic acid	[77]
	twigs	Powder samples	HPLC-DAD	Prenylated flavonoids: kuwanon G; morusin; Stilbenes: mulberroside A; oxyresveratrol Phenolic acid:4-hydroxycinnamic acid	[77]
	root bark	Powder bark samples	HPLC-DAD	Flavanonol: taxifolin Prenylated flavonoids: kuwanon G; morusin; Stilbenes: mulberroside A; oxyresveratrol Phenolic acid: chlorogenic acid;	[77]
		Dried root bark	MPLC	Prenylated flavonoid: kuwanon G Diels–Alder-type adducts: mulberrofuran G; albanol B	[78]

UPLC-TUV/QDa—high-performance liquid chromatography with tunable ultraviolet and quadrupole dalton detectors; HPLC-DAD-ESI-HRMS—ultra-high-performance liquid chromatography diode array detector electrospray ionisation high-resolution mass spectrometry; HPLC—high-performance liquid chromatography; LC-MS-MS —liquid chromatography with tandem mass spectrometry (LC-MS-MS); LC-MS—reversed-phase coupled to high-resolution mass spectrometry; RP-MPLC—reversed-phase medium-pressure liquid chromatography; MS—mass spectrometry; HPLC-MS—high-performance liquid chromatography-mass spectrometry; HPLC-DAD—high-performance liquid chromatography with diode array detection; MPLC—medium-pressure preparative liquid chromatography.

**Table 3 plants-11-00152-t003:** The in vitro SET assays that investigated antioxidant capacity of mulberry fruits and leaves from data from the literature (2016–2021).

Species	Organs	ABTS	DPPH	FRAP	References
*Morus nigra*	fruits	Not recorded	Not recorded	21.33 ± 0.35 µmol TE/g dw	[49]
		600.31 µmol TE/L	131.27 µmol TE/L	Not recorded	[114]
		5.842 ± 0.1155 mmol TE/L	46.94 ± 1.68%	0.4627 ± 0.0101 mmol TE/L	[115]
		6.43 mg VCE/g fw	2.51 mg VCE/g fw	Not recorded	[53]
	leaf	21.85 mg TE/g dw	146.04 mg TE/g dw	52.71 mg TE/g dw	[54]
		9.89 ± 0.87 mM TE	Not recorded	Not recorded	[40]
*Morus atropur purea Roxb.*	fruits	4.11 ± 0.48 µg/mL (IC 50)	10.08 ± 1.12 µg/mL (IC 50)	Not recorded	[55]
*Morus alba*	fruits	92.15 g TE/100 g dw	10.70 g TE/100 g dw	Not recorded	[116]
		0.52 mg VCE/g fw	0.21 mg VCE/g fw	Not recorded	[53]
		Not recorded	5.85 to 40.73 mg TE/g dw	1.33 to 82.87 mg TE/g dw	[56]
		19.37 ± 3.67 µg/mL (EC 50)	38.31 ± 2.13 µg/mL (EC50)	1.23 µM Fe^2+^	[57]
	leaf	23.63 ± 0.019 µM TE/g	49.42 ± 0.005 µM TE/g	0.0221 ± 0.042 µM Fe^2+^/mg	[58]
		4.47 ± 0.20 mg/mL (IC 50)	2.95 ± 0.66 mg/mL (IC 50)	Not recorded	[59]
		6.12 ± 0.53 mM TE	Not recorded	Not recorded	[40]
	twig	92.15 g TE/100 g dw	10.70 g TE/100 g dw	Not recorded	[116]

DPPH—2–2 diphenyl-1-picrylhydrazyl assay, FRAP—ferric reducing ability of plasma assay, ABTS—2,2′-azinobis-(3-ethylbenzothiazoline-6-sulfonic acid) assay, TE—Trolox Equivalent; VCE—vitamin C equivalent.

**Table 4 plants-11-00152-t004:** The health-promoting properties of different parts of mulberry fruits, as determined in various experimental designs (animal and cell models) (2017–2021).

Health Effect	Part	Species	Sample Type	Experimental Model	Main Outcomes	References
Anti-inflammatory effect	fruits	*M. nigra*	Aqueous extract	Male Wistar rats, periodontal soft tissues	Decreased MMP-8 and MMP-13 levels in periodontal tissue. Inhibited alveolar bone resorption by suppressing the expression of RANKL and OPG.	[162]
Anti-Parkinson effect		*M. nigra*	Mulberry juice (cyanidin-3-glucoside, 137 mg/100 g)	LID in MPTP MPTP)-induced PD in male BALB/c mice	Mulberry juice (10–15 mL/kg) for one week may be effective for controlling LID in MPTP-induced PD.	[107]
		*M. alba*	Lyophilized mulberry extract	MPTP/p model of early PD in male mice C57BL/6	Improved PD-related, non-motor symptoms by inhibiting olfactory dysfunction and motor deficits. The protective effects against dopaminergic neuronal damage induced by MPTP/p in the substantia nigra and striatum. Inhibited the up-regulation of α-synuclein and ubiquitin.	[163]
Cardiovascular effect/Antiatherosclerosis		*M. nigra*	Ethanolic extract	Rats, Sprague–Dawley	Significantly decreased the content of malondialdehyde and improved the anti-oxidative enzymatic activities, attenuated hepatic steatosis, reduced intima-media thickness and suppressed the development of arterial atherosclerosis by regulating lipid metabolism abnormalities, strengthening anti-oxidant activities and reducing atherosclerotic lesions.	[106]
Hepatoprotective effect		*M. nigra*	Aqueous extracts.	Human hepatocellular carcinoma (HepG2)-three concentrations (0.01, 0.1 and 1 mg/mL) and compared to silymarin	Black mulberries; phenolic compounds are beneficial for counteracting liver toxins.	[157]
Gastroprotective effect		*M. nigra*	Methanolic extract	Female Swiss mice	Effective defense of the gastric mucosa against the acidified methanol, only at 300 mg/kg (p.o.), reducing the ulcer area by 64.06%.	[164]
Antinociceptive effect		*M. nigra* and *M. alba*	Mulberry dry powder	Male Kunming mice, three main flavonoids tested (C3G, Ru and IQ)	Neither C3G, Ru nor IQ individually reduced the duration of both phases, while the mix (C3G, Ru and IQ) significantly reduced the duration of the secondary phase (inflammatory pain phase).	[165]
Antibacterial effect		*M. nigra* and *M. alba*	Mulberry dry powder of *M. nigra*, *M. mongolica* and *M. alba* ‘Zhenzhubai’	*E. coli*, *S. aureus*, *P. aeruginosa*	*M. nigra* presented stronger inhibitory activity against *S. aureus* compared with *E. coli* in the MBC test.	[165]
Antimicrobial effect	seed	*M. nigra*	Hydroethanolic extracts lyophilized	Six Gram-negative bacteria, three Gram-positive bacteria and one yeast	Efficacy of mulberry extract was shown against *S. aureus*, resistant to methicillin with MIC values of 5 mg/mL.	[166]
Cytotoxic effect		*M. nigra*	Hydroethanolic extracts lyophilized	Human tumor cell lines: MCF-7, HepG2 NCI-H460 and HeLa cells. Primary cellular line from porcine liver as normal cell line	The mulberry extract at concentration of 400 µg/mL was not effective against tumor and normal cells.	[166]
Treating climacteric symptoms	leaves	*M. nigra*	Leaves powder	62 climacteric women	Climacteric symptoms and quality of life analysis (functional capacity, vitality, mental health and social aspect) were improved after administration of 250 mg of *M. nigra* leaves powder for 60 days.	[167]
Antidepressant and neuroprotective effects		*M. nigra*	Aqueous extract	Ex vivo and in vitro model in male Swiss mice, gavage administration	Treatment with *M. nigra* (3 mg/kg) decreased the immobility time in the TST. *M. nigra* extract protect hippocampal and cerebrocortical slices against glutamate-induced damage via PI3K/Akt/GSK-3β pathway.	[86]
Anti-Melanogenesis effect		*M. alba*	Dried leaves	B16-F10 mouse skin melanoma cells	*M. alba* can be an excellent natural source for skin-whitening agents. Moracin J identified in the leaves, decreased melanin production and intracellular tyrosinase activity by modulating CREB and p38 signaling pathways in the cells of melanoma B16-F10 activated α-MSH.	[168]
α-Glucosidase inhibition and tyrosinase inhibitory	twigs	*M. nigra*	Dried powder	Ethanol extract portioned in 6 fractions	Among13 compounds isolated from the twigs, Nigranol B and sanggenol H exhibited powerful α-glucosidase inhibitory activities with IC50 values at 1.63 and 1.43 µM, respectively.	[169]
Anti-hyperuricemic effect		*M. alba*	Mori ramulus refined extract (ZY1402-A)	Adult male SPF Kunming mice, intragastric administration	The extract significantly reduced the serum uric acid levels of SPF Kunming mice.	[170]
Antileukemic effect	bark	*M. nigra*	Bark meal	Parental Jurkat A3 leukemia cell line; FADD-deficient Jurkat Cells; caspase 9-deficient Jurkat cells; Caspase 8- and 10-doubly deficient Jurkat cells	Morniga-G activates T, B, and NK Lymphocytes and induces the cell death of Tn-positive leukemia lymphocytes. cells via concomitant O-glycosylation, caspase and TRAIL/DR5-dependent pathways.	[155]
Vascular protective effect		*M. alba*	Ethanolic extract	Male rats (Sprague–Dawley)	Potent endothelium-dependent vasodilator through endothelial-dependent NO-cGMP pathway, including the activation of TEA sensitive K+ channels. White mulberry extract attenuated PDGF-BB induced VSMCs proliferation and migration.	[171]
Relaxant effect						

MM-8 = Matrix metalloproteinase-8; MM-13 = Matrix metalloproteinase-13; RANKL—receptor activation of nuclear factor κB (RANK) ligand; OPG—osteoprotegerin; MPTP—1-methyl-4-phenyl-1,2,3,6-tetrahydropyridine; LID—levodopa-induced dyskinesia; MBC—minimum bactericidal concentration; C3G—cyanidin 3-O-glucoside; TST—tail suspension test; Ru—rutinoside; IQ—isoquercetin; MCF-7—mammary adenocarcinoma; HepG2 cells—human hepatocellular carcinoma; NCI-H460—small cell carcinoma; HeLa—cervical carcinoma; QoL—quality of life analysis; MRSA—*Staphylococcus aureus*, resistant to methicillin; MIC—minimal inhibitory concentrations; SRB—sulforodamine B; PD—Parkinson’s disease; NO-cGMP—nitric oxide cyclic-guanosine monophosphate; TEA—tetraethylammonium; K+—potassium; PDGF—platelet-derived growth factors; VSMCs—vascular smooth muscle cells.

**Table 5 plants-11-00152-t005:** Product foodstuffs, their compositions and health benefits (2016–2021).

Product Foodstuff	Major Findings	Reference
Black mulberry food colorants	Three formulations of solid natural colorants based on black mulberry anthocyanins (cyanidin-3-O-glucoside and cyanidin-O-rhamnoside), obtained through the spray-drying technique, were developed. These natural additives have a good stability in time and a variation of anthocyanin content and color parameters during the 12 weeks of storage, at room and refrigerated temperatures.	[176]
Mulberry gummy candies	Gummy candies obtained from 5, 7.5 and 10 g of mulberry molasses/100 g gelatin illustrate the potential for using molasses in a healthier development of confectionery products. These candies contain natural sugars, thus replacing sugar syrup or artificial sweeteners.	[178]
Mulberry leaf powder drink	The effect on adults of consuming of biscuits with a beverage of powdered mulberry leaves in the afternoon on postprandial glucose levels at dinner was a significant reduction in postprandial increases in glucose.	[182]
Mulberry leaf tea	The quercetin 3-O-malonylglucoside and kaempferol 3-O-malonylglucoside found in white mulberry leaves can be used as ingredients for a functional food to improve the health benefits, such as controlling blood glucose, preventing aging-related diseases and regulating glycolipid metabolic abnormalities.	[183,187,188]
Black mulberry dietary syrup	Administered in different concentrations in the diet of fish, the syrup, increased activities of serum lysozyme, myeloperoxidase, superoxide dismutase and catalase, and increased the expression levels of immune-related genes in the spleen and antioxidant-related genes in the liver of fish fed.	[181]
Rapeseed honey with mulberry leaves and fruits	The addition of dried leaves and freeze-dried fruits (4%, w/v) to rapeseed honey added value to the product by increasing the content of flavonoids and phenolic acids and antioxidant capacity.	[184]
Black mulberry-aged wines	The non-thermal processing applied at wine maturation point can be a potential method of improving the maturation process by modifying the chromatic properties of the wine.	[179]
	In the volatile composition of the non-thermal, accelerated, aged wines, many volatile compounds were found that are grouped into nine chemical families: alcohols (32), esters (53), acids (14), volatile phenols (11), aldehydes (16), ketones (15), terpenes (11), lactones (11) and furans (3).	[180]
Black mulberry jam	Black mulberries were processed into jam on an industrialized scale. The total phenols, flavonoids, anthocyanins and antioxidant capacity was significantly decreased but % recovery of bioaccessible the natural compounds increased after jam processing.	[177]
Dark chocolate with black mulberry	Dark chocolate was fortified with dry black mulberry waste extract, encapsulated in chitosan-coated liposomes. This formula was shown to protect the anthocyanin content and increase the bioavailability of these pigments in vitro.	[186]

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
