# Peer review of "Phytochemical Composition of Different Botanical Parts of Morus Species, Health Benefits and Application in Food Industry"

_plants, 2022, doi:10.3390/plants11020152_

Round 1
Reviewer 1 Report
Manuscript is interesting and summarizes the information on Morus species. It is quite well written. However, I have some suggestions for Authors:
- Line 141-142: the content of vit C is given in mg/100 ml. Is it correct? In this form, the information is useless because there is no mention how the solution was prepared (e.g. how many plant material was taken for extraction)
- Line 206: “In generally, the bioactive compounds in M. nigra L. are higher compared to M. alba L.” – not precise statement. Has higher content of bioactive compounds?
- Line 251: “effects of solvent concentration” – what solvent?
- Line 253: it should be: “depends on”
- In my opinion, Fig.2 is unnecessary and may be moved to Supplementary material. The compounds are not new and the structure is known.
- Line 310: use full name for “cy-3-glu “
- Table 2: “Identifier system” – the general term “technique” is better because e.g. MPLC was used for separation (type of detection is not mentioned).
“Anthocyanins: Cyanidin (Cyanidin…)” – it is misleading because it is not clear whether aglycone form (Cyanidin ) was present in plant or only its derivatives specified in bracket. Remove bracket or add “Cyanidin derivatives” – the same comment for Kaempferol, quercetin etc…
Add explanation for “CC” under the Table
- Lines 404-439: In my opinion, this part should be shortened. There are a lot of general information here not linked directly with the aim of the study, e.g. mechanisms of antioxidant tests is unnecessary.
- Table 3: remove CUPRAC from the table because this type of activity was not recorded.
- Line 467: it should be: “enhanced”
- Line 479: what does it mean “ps”?
- Generally 4.1. section is chaotic a little bit. In this section the other types of activity are also described (e.g. lines 486-490)
- Line 583/590: what extract? Methanolic?
- References: abbreviation should be consequently used for the names of the journals
- There are some editorial errors: e.g. unnecessary dots (line 201, 340, 392… ), line 248: GGAE, unnecessary capital letters for the name of compounds (e.g. line 326,327, 334….)
Author Response
Response to reviewer #1
Thanks for your suggestions that improve our article.
- Line 141-142: the content of vit C is given in mg/100 ml. Is it correct? In this form, the information is useless because there is no mention how the solution was prepared (e.g. how many plant material was taken for extraction)
Response: According to the reference [26], a total of 30 mulberry fruits were taken for each replication and Ascorbic acid was quantified with the reflectometer (Reflectometer RQflex®, Merck KGaA, Darmstadt, Germany) and the results were expressed as mg of ascorbic acid (AsA) per 100 mL.
I gave the details about the quantification of vitamin C according to the reference 26 in the manuscript, with the red colour.
- Line 206: “In generally, the bioactive compounds in M. nigra L. are higher compared to M. alba L.” – not precise statement. Has higher content of bioactive compounds?
Response: You are right. I completed the phrase: “In generally, the content of bioactive compounds in M. nigra L. are higher compared to M. alba L. due to the presence of anthocyanins [29.55]”.
- Line 251: “effects of solvent concentration” – what solvent?
Response: I completed in manuscript with red colour the kind of solvent (ethanol) and the concentration used.
- Line 253: it should be: “depends on”
Response: Thank you. I corrected and marked with red colour.
- In my opinion, Fig.2 is unnecessary and may be moved to Supplementary material. The compounds are not new and the structure is known.
Response: Thank you very much for your suggestion. The Fig. 2 is moved to Supplementary material.
- Line 310: use full name for “cy-3-glu “
Response: I included the full name : cyanidin-3-glucoside.
- Table 2: “Identifier system” – the general term “technique” is better because e.g. MPLC was used for separation (type of detection is not mentioned).
Response: I replaced the “Identifier system” with “Technique”. Thanks for your suggestion.
“Anthocyanins: Cyanidin (Cyanidin…)” – it is misleading because it is not clear whether aglycone form (Cyanidin ) was present in plant or only its derivatives specified in bracket. Remove bracket or add “Cyanidin derivatives” – the same comment for Kaempferol, quercetin etc…
Response: Thanks for your suggestion. I corrected in the Table 2 according your suggestion and we chose to remove brackets.
Add explanation for “CC” under the Table
Response: I removed CC (column chromatography)- it was an error.
- Lines 404-439: In my opinion, this part should be shortened. There are a lot of general information here not linked directly with the aim of the study, e.g. mechanisms of antioxidant tests is unnecessary.
Response: According to your suggestion the text was shorted.
- Table 3: remove CUPRAC from the table because this type of activity was not recorded.
Response: We removed CUPRAC from the table and from legend.
- Line 467: it should be: “enhanced”
Response: We changed it.
- Line 479: what does it mean “ps”?
Response: It is our mistake. The correct is ‘dw’. I corrected. Thanks.
- Generally 4.1. section is chaotic a little bit. In this section the other types of activity are also described (e.g. lines 486-490)
Response: You're right. I removed that section.
- Line 583/590: what extract? Methanolic?
Response: We specified the kind of extract in both cases with red colour in the manuscript.
- References: abbreviation should be consequently used for the names of the journals
Response: We checked and corrected the references
- There are some editorial errors: e.g. unnecessary dots (line 201, 340, 392… ), line 248: GGAE, unnecessary capital letters for the name of compounds (e.g. line 326,327, 334….)
Response: We corrected the editorial errors and the names of the compounds in lower case were modified. Thank you very much for your comments.

Reviewer 2 Report
In my opinion:
- Particle patterns (Fg.2.A) should be converted to Haworth projections. Those posted are illegible. It is difficult to define even alpha or beta configurations.
- Please don't get me wrong but:
- Research work is based on facts. All the studies cited in this articles should prove the scientifically beneficial effect of a substance isolated from the tested plant on human health.
- Authors should avoid works and citing studies where conclusions are formulated as potential properties, potential action, potential application, etc. We are well aware that such research does bring not to much.
- Authors should once again decide whether they want to cite scientifically proven evidence or presumptions.
- The work is done correctly and with a logical division. The content of the article is still valid. From the beginning of human existence, plants served as medicines and food. Now pharmacy is back to basics again. The work is interesting and synthetically written.
Author Response
Response to reviewer #2
- Particle patterns (Fg.2.A) should be converted to Haworth projections. Those posted are illegible. It is difficult to define even alpha or beta configurations.
The Figure 2 was moved to the supplementary files and we converted the 3D structure in Haworth projections.
2.Please don't get me wrong but:
- Research work is based on facts. All the studies cited in this articles should prove the scientifically beneficial effect of a substance isolated from the tested plant on human health.
- Authors should avoid works and citing studies where conclusions are formulated as potential properties, potential action, potential application, etc. We are well aware that such research does bring not to much.
- Authors should once again decide whether they want to cite scientifically proven evidence or presumptions.
We designed this review based on the idea that each part of the mulberry has a specific phytochemical footprint and implicitly characteristic biological effects. Research studies are focused on extracts / compounds individual from fruits or leaves or roots and their beneficial effects. Through this article I wanted to provide an overview of all the organs of the mulberry both in terms of chemical composition and beneficial effects and in addition its applicability in the food industry. Given that the name of the special volume is "Medicinal Plant extract", I focused on various mulberry extracts, which are indeed a "cocktail" of phytocompounds with potential beneficial effects in various diseases.
3.The work is done correctly and with a logical division. The content of the article is still valid. From the beginning of human existence, plants served as medicines and food. Now pharmacy is back to basics again. The work is interesting and synthetically written.
Thank you very much for your appreciation.
But you're right, studies that show various biological effects based on compounds isolated / purified from plants provide valuable information. Based on your suggestion, in the future we will address a topic based on the compounds isolated from plants and their concrete effects or we will try to highlight various synergistic / additional or even antagonistic effects between various compounds present in plants.

Reviewer 3 Report
Dear Authors,
The work entitled: "Phytochemical composition of various botanical parts of the Morus species, health benefits and application in the food industry" due to the nutritional value, phytochemical properties, and its beneficial effect on human health, including antioxidant, anticancer, immunomodulating and antidiabetic effects, deserves publication. However, the authors did not avoid mistakes.
Here are the detailed notes:
- The purpose of the work should be clearer.
- The work should contain an alternative research hypothesis, set against the null hypothesis
- In the introduction, there is a mistake in the fourth reference to literature
- There is no chapter "Research methodology", in which Authors should explain the essence of scientific cognition, research processes, research stages and research procedure. This chapter should also include the types and forms of scientific work as well as guidelines for the implementation of the concept of scientific work, present methods, techniques, and research tools, including the activities of optimal research.
- The work structure is messy. It should be improved, e.g., the discussion of the importance of the nutritional value of mulberry is inconsistent, the content of minerals is discussed at the beginning and end of the section "fruits". This section is discussed a second time later in the manuscript, can't this information be cumulative?
- Detailed data on the content of individual elements and other chemical components of the fruit of Morus species should be included in the table, with reference to the most recent sources.
- There is no discussion or summary.

Author Response
Response to reviewer #3
Thanks for your suggestions that improve our article.
- The purpose of the work should be clearer.
The aim of the work was highlighted in the introduction section.
- The work should contain an alternative research hypothesis, set against the null hypothesis
As our work was based on literature research, the PRISMA diagram was added in order to show a transparent criteria of literature selection. Alternative research hypothesis, set against the null hypothesis has not been considered in this paper, but is will be considered for our next project, including original research data.
- In the introduction, there is a mistake in the fourth reference to literature
The correction was applied.
- There is no chapter "Research methodology", in which Authors should explain the essence of scientific cognition, research processes, research stages and research procedure. This chapter should also include the types and forms of scientific work as well as guidelines for the implementation of the concept of scientific work, present methods, techniques, and research tools, including the activities of optimal research.
According to your suggestion, a new section was added, Research methodology, including PRISMA flow diagram. Thank you very much for your suggestion.
- The work structure is messy. It should be improved, e.g., the discussion of the importance of the nutritional value of mulberry is inconsistent, the content of minerals is discussed at the beginning and end of the section "fruits". This section is discussed a second time later in the manuscript, can't this information be cumulative?
According to your suggestions, the discussion section was reconsidered and improved.
- Detailed data on the content of individual elements and other chemical components of the fruit of Morus species should be included in the table, with reference to the most recent sources.
According to your suggestions, table 2 has been reconsidered and additional information added.
- There is no discussion or summary.
The discussion section has been reconsidered.
